# InfoGAIL: Interpretable Imitation Learning from Visual Demonstrations

**Yunzhu Li**
MIT
liyunzhu@mit.edu

**Jiaming Song**
Stanford University
tsong@cs.stanford.edu

**Stefano Ermon**
Stanford University
ermon@cs.stanford.edu

## Abstract

The goal of imitation learning is to mimic expert behavior without access to an explicit reward signal. Expert demonstrations provided by humans, however, often show significant variability due to latent factors that are typically not explicitly modeled. In this paper, we propose a new algorithm that can infer the latent structure of expert demonstrations in an unsupervised way. Our method, built on top of Generative Adversarial Imitation Learning, can not only imitate complex behaviors, but also learn interpretable and meaningful representations of complex behavioral data, including visual demonstrations. In the driving domain, we show that a model learned from human demonstrations is able to both accurately reproduce a variety of behaviors and accurately anticipate human actions using raw visual inputs. Compared with various baselines, our method can better capture the latent structure underlying expert demonstrations, often recovering semantically meaningful factors of variation in the data.

## 1 Introduction

A key limitation of reinforcement learning (RL) is that it involves the optimization of a predefined reward function or reinforcement signal [1–6]. Explicitly defining a reward function is straightforward in some cases, e.g., in games such as Go or chess. However, designing an appropriate reward function can be difficult in more complex and less well-specified environments, e.g., for autonomous driving where there is a need to balance safety, comfort, and efficiency.

Imitation learning methods have the potential to close this gap by learning how to perform tasks directly from expert demonstrations, and has succeeded in a wide range of problems [7–11]. Among them, Generative Adversarial Imitation Learning (GAIL, [12]) is a model-free imitation learning method that is highly effective and scales to relatively high dimensional environments. The training process of GAIL can be thought of as building a generative model, which is a stochastic policy that when coupled with a fixed simulation environment, produces similar behaviors to the expert demonstrations. Similarity is achieved by jointly training a discriminator to distinguish expert trajectories from ones produced by the learned policy, as in GANs [13].

In imitation learning, example demonstrations are typically provided by human experts. These demonstrations can show significant variability. For example, they might be collected from multiple experts, each employing a different policy. External latent factors of variation that are not explicitly captured by the simulation environment can also significantly affect the observed behavior. For example, expert demonstrations might be collected from users with different skills and habits. The goal of this paper is to develop an imitation learning framework that is able to automatically discover and disentangle the latent factors of variation underlying expert demonstrations. Analogous to the goal of uncovering style, shape, and color in generative modeling of images [14], we aim to automatically learn similar interpretable concepts from human demonstrations through an unsupervised manner.

We propose a new method for learning a latent variable generative model that can produce trajectories in a dynamic environment, i.e., sequences of state-actions pairs in a Markov Decision Process. Not only can the model accurately reproduce expert behavior, but also empirically learns a latent space of the observations that is semantically meaningful. Our approach is an extension of GAIL, where the objective is augmented with a mutual information term between the latent variables and the observed state-action pairs. We first illustrate the core concepts in a synthetic 2D example and then demonstrate an application in autonomous driving, where we learn to imitate complex driving behaviors while recovering semantically meaningful structure, without any supervision beyond the expert trajectories. [1] Remarkably, our method performs directly on raw visual inputs, using raw pixels as the only source of perceptual information. The code for reproducing the experiments are available at `https://github.com/ermongroup/InfoGAIL`.

In particular, the contributions of this paper are threefold:

1. We extend GAIL with a component which approximately maximizes the mutual information between latent space and trajectories, similar to InfoGAN [14], resulting in a policy where low-level actions can be controlled through more abstract, high-level latent variables.

2. We extend GAIL to use raw pixels as input and produce human-like behaviors in complex high-dimensional dynamic environments.

3. We demonstrate an application to autonomous highway driving using the TORCS driving simulator [15]. We first demonstrate that the learned policy is able to correctly navigate the track without collisions. Then, we show that our model learns to reproduce different kinds of human-like driving behaviors by exploring the latent variable space.

## 2 Background

### 2.1 Preliminaries

We use the tuple $(\mathcal{S}, \mathcal{A}, P, r, \rho_0, \gamma)$ to define an infinite-horizon, discounted Markov decision process (MDP), where $\mathcal{S}$ represents the state space, $\mathcal{A}$ represents the action space, $P : \mathcal{S} \times \mathcal{A} \times \mathcal{S} \to \mathbb{R}$ denotes the transition probability distribution, $r : \mathcal{S} \to \mathbb{R}$ denotes the reward function, $\rho_0 : \mathcal{S} \to \mathbb{R}$ is the distribution of the initial state $s_0$, and $\gamma \in (0, 1)$ is the discount factor. Let $\pi$ denote a stochastic policy $\pi : \mathcal{S} \times \mathcal{A} \to [0, 1]$, and $\pi_E$ denote the expert policy to which we only have access to demonstrations. The expert demonstrations $\tau_E$ are a set of trajectories generated using policy $\pi_E$, each of which consists of a sequence of state-action pairs. We use an expectation with respect to a policy $\pi$ to denote an expectation with respect to the trajectories it generates: $\mathbb{E}_\pi[f(s, a)] \triangleq \mathbb{E}[\sum_{t=0}^{\infty} \gamma^t f(s_t, a_t)]$, where $s_0 \sim \rho_0$, $a_t \sim \pi(a_t|s_t)$, $s_{t+1} \sim P(s_{t+1}|a_t, s_t)$.

### 2.2 Imitation learning

The goal of imitation learning is to learn how to perform a task directly from expert demonstrations, without any access to the reinforcement signal $r$. Typically, there are two approaches to imitation learning: 1) behavior cloning (BC), which learns a policy through supervised learning over the state-action pairs from the expert trajectories [16]; and 2) apprenticeship learning (AL), which assumes the expert policy is optimal under some unknown reward and learns a policy by recovering the reward and solving the corresponding planning problem. BC tends to have poor generalization properties due to compounding errors and covariate shift [17, 18]. AL, on the other hand, has the advantage of learning a reward function that can be used to score trajectories [19–21], but is typically expensive to run because it requires solving a reinforcement learning (RL) problem inside a learning loop.

### 2.3 Generative Adversarial Imitation Learning

Recent work on AL has adopted a different approach by learning a policy without directly estimating the corresponding reward function. In particular, Generative Adversarial Imitation Learning (GAIL, [12]) is a recent AL method inspired by Generative Adversarial Networks (GAN, [13]). In the GAIL framework, the agent imitates the behavior of an expert policy $\pi_E$ by matching the generated state-action distribution with the expert's distribution, where the optimum is achieved when the

distance between these two distributions is minimized as measured by Jensen-Shannon divergence. The formal GAIL objective is denoted as

$$\min_{\pi} \max_{D \in (0,1)^{S \times A}} \mathbb{E}_{\pi}[\log D(s,a)] + \mathbb{E}_{\pi_E}[\log(1 - D(s,a))] - \lambda H(\pi) \tag{1}$$

where $\pi$ is the policy that we wish to imitate $\pi_E$ with, $D$ is a discriminative classifier which tries to distinguish state-action pairs from the trajectories generated by $\pi$ and $\pi_E$, and $H(\pi) \triangleq \mathbb{E}_{\pi}[-\log \pi(a|s)]$ is the $\gamma$-discounted causal entropy of the policy $\pi_{\theta}$ [22]. Instead of directly learning a reward function, GAIL relies on the discriminator to guide $\pi$ into imitating the expert policy.

GAIL is model-free: it requires interaction with the environment to generate rollouts, but it does not need to construct a model for the environment. Unlike GANs, GAIL considers the environment/simulator as a black box, and thus the objective is not differentiable end-to-end. Hence, optimization of GAIL objective requires RL techniques based on Monte-Carlo estimation of policy gradients. Optimization over the GAIL objective is performed by alternating between a gradient step to increase (1) with respect to the discriminator parameters, and a Trust Region Policy Optimization (TRPO, [2]) step to decrease (1) with respect to $\pi$.

## 3 Interpretable Imitation Learning through Visual Inputs

Demonstrations are typically collected from human experts. The resulting trajectories can show significant variability among different individuals due to internal latent factors of variation, such as levels of expertise and preferences for different strategies. Even the same individual might make different decisions while encountering the same situation, potentially resulting in demonstrations generated from multiple near-optimal but distinct policies. In this section, we propose an approach that can 1) discover and disentangle salient latent factors of variation underlying expert demonstrations without supervision, 2) learn policies that produce trajectories which correspond to these latent factors, and 3) use visual inputs as the only external perceptual information.

Formally, we assume that the expert policy is a mixture of experts $\pi_E = \{\pi_E^0, \pi_E^1, \dots\}$, and we define the generative process of the expert trajectory $\tau_E$ as: $s_0 \sim \rho_0$, $c \sim p(c)$, $\pi \sim p(\pi|c)$, $a_t \sim \pi(a_t|s_t)$, $s_{t+1} \sim P(s_{t+1}|a_t, s_t)$, where $c$ is a discrete latent variable that selects a specific policy $\pi$ from the mixture of expert policies through $p(\pi|c)$ (which is unknown and needs to be learned), and $p(c)$ is the prior distribution of $c$ (which is assumed to be known before training). Similar to the GAIL setting, we consider the apprenticeship learning problem as the dual of an occupancy measure matching problem, and treat the trajectory $\tau_E$ as a set of state-action pairs. Instead of learning a policy solely based on the current state, we extend it to include an explicit dependence on the latent variable $c$. The objective is to recover a policy $\pi(a|s, c)$ as an approximation of $\pi_E$; when $c$ is samples from the prior $p(c)$, the trajectories $\tau$ generated by the conditional policy $\pi(a|s, c)$ should be similar to the expert trajectories $\tau_E$, as measured by a discriminative classifier.

### 3.1 Interpretable Imitation Learning

Learning from demonstrations generated by a mixture of experts is challenging as we have no access to the policies employed by the individual experts. We have to proceed in an unsupervised way, similar to clustering. The original Generative Adversarial Imitation Learning method would fail as it assumes all the demonstrations come from a single expert, and there is no incentive in separating and disentangling variations observed in the data. A method that can automatically disentangle the demonstrations in a meaningful way is thus needed.

The way we address this problem is to introduce a latent variable $c$ into our policy function, $\pi(a|s, c)$. Without further constraints over $c$, applying GAIL directly to this $\pi(a|s, c)$ could simply ignore $c$ and fail to separate different types of behaviors present in the expert trajectories [2]. To incentivize the model to use $c$ as much as possible, we utilize an information-theoretic regularization enforcing that there should be high mutual information between $c$ and the state-action pairs in the generated trajectory. This concept was introduced by InfoGAN [14], where latent codes are utilized to discover the salient semantic features of the data distribution and guide the generating process. In particular, the regularization seeks to maximize the mutual information between latent codes and trajectories,

denoted as $I(c; \tau)$,which is hard to maximize directly as it requires access to the posterior $P(c|\tau)$. Hence we introduce a variational lower bound, $L_I(\pi, Q)$, of the mutual information $I(c; \tau)$[3]:

$$
\begin{aligned}
L_I(\pi, Q) &= \mathbb{E}_{c \sim p(c), a \sim \pi(\cdot|s,c)}[\log Q(c|\tau)] + H(c) \\
&\leq I(c; \tau)
\end{aligned}
\tag{2}
$$

where $Q(c|\tau)$ is an approximation of the true posterior $P(c|\tau)$. The objective under this regularization, which we call Information Maximizing Generative Adversarial Imitation Learning (InfoGAIL), then becomes:

$$
\min_{\pi, Q} \max_D \mathbb{E}_\pi[\log D(s,a)] + \mathbb{E}_{\pi_E}[\log(1 - D(s,a))] - \lambda_1 L_I(\pi, Q) - \lambda_2 H(\pi)
\tag{3}
$$

where $\lambda_1 > 0$ is the hyperparameter for information maximization regularization term, and $\lambda_2 > 0$ is the hyperparameter for the casual entropy term. By introducing the latent code, InfoGAIL is able to identify the salient factors in the expert trajectories through mutual information maximization, and imitate the corresponding expert policy through generative adversarial training. This allows us to disentangle trajectories that may arise from a mixture of experts, such as different individuals performing the same task.

To optimize the objective, we use a simplified posterior approximation $Q(c|s, a)$, since directly working with entire trajectories $\tau$ would be too expensive, especially when the dimension of the observations is very high (such as images). We then parameterize policy $\pi$, discriminator $D$ and posterior approximation $Q$ with weights $\theta$, $\omega$ and $\psi$ respectively. We optimize $L_I(\pi_\theta, Q_\psi)$ with stochastic gradient methods, $\pi_\theta$ using TRPO [2], and $Q_\psi$ is updated using the Adam optimizer [23]. An outline for the optimization procedure is shown in Algorithm 1.

---

**Algorithm 1** InfoGAIL

---

**Input:** Initial parameters of policy, discriminator and posterior approximation $\theta_0, \omega_0, \psi_0$; expert trajectories $\tau_E \sim \pi_E$ containing state-action pairs.
**Output:** Learned policy $\pi_\theta$
**for** $i = 0, 1, 2, ...$ **do**
    Sample a batch of latent codes: $c_i \sim p(c)$
    Sample trajectories: $\tau_i \sim \pi_{\theta_i}(c_i)$, with the latent code fixed during each rollout.
    Sample state-action pairs $\chi_i \sim \tau_i$ and $\chi_E \sim \tau_E$ with same batch size.
    Update $\omega_i$ to $\omega_{i+1}$ by ascending with gradients

$$
\Delta_{\omega_i} = \hat{\mathbb{E}}_{\chi_i}[\nabla_{\omega_i} \log D_{\omega_i}(s,a)] + \hat{\mathbb{E}}_{\chi_E}[\nabla_{\omega_i} \log(1 - D_{\omega_i}(s,a))]
$$

    Update $\psi_i$ to $\psi_{i+1}$ by descending with gradients

$$
\Delta_{\psi_i} = -\lambda_1 \hat{\mathbb{E}}_{\chi_i}[\nabla_{\psi_i} \log Q_{\psi_i}(c|s,a)]
$$

    Take a policy step from $\theta_i$ to $\theta_{i+1}$, using the TRPO update rule with the following objective:

$$
\hat{\mathbb{E}}_{\chi_i}[\log D_{\omega_{i+1}}(s,a)] - \lambda_1 L_I(\pi_{\theta_i}, Q_{\psi_{i+1}}) - \lambda_2 H(\pi_{\theta_i})
$$

**end for**

---

### 3.2 Reward Augmentation

In complex and less well-specified environments, imitation learning methods have the potential to perform better than reinforcement learning methods as they do not require manual specification of an appropriate reward function. However, if the expert is performing sub-optimally, then any policy trained under the recovered rewards will be also suboptimal; in other words, the imitation learning agent's potential is bounded by the capabilities of the expert that produced the training data. In many cases, while it is very difficult to fully specify a suitable reward function for a given task, it is relatively straightforward to come up with constraints that we would like to enforce over the policy.

This motivates the introduction of *reward augmentation* [8], a general framework to incorporate prior knowledge in imitation learning by providing additional incentives to the agent without interfering

with the imitation learning process. We achieve this by specifying a surrogate state-based reward $\eta(\pi_\theta) = \mathbb{E}_{s \sim \pi_\theta}[r(s)]$ that reflects our bias over the desired agent's behavior:

$$\min_{\theta,\psi} \max_\omega \mathbb{E}_{\pi_\theta}[\log D_\omega(s,a)] + \mathbb{E}_{\pi_E}[\log(1 - D_\omega(s,a))] - \lambda_0 \eta(\pi_\theta) - \lambda_1 L_I(\pi_\theta, Q_\psi) - \lambda_2 H(\pi_\theta) \quad (4)$$

where $\lambda_0 > 0$ is a hyperparameter. This approach can be seen as a hybrid between imitation and reinforcement learning, where part of the reinforcement signal for the policy optimization is coming from the surrogate reward and part from the discriminator, i.e., from mimicking the expert. For example, in our autonomous driving experiment below we show that by providing the agent with a penalty if it collides with other cars or drives off the road, we are able to significantly improve the average rollout distance of the learned policy.

### 3.3 Improved Optimization

While GAIL is successful in tasks with low-dimensional inputs (in [12], the largest observation has 376 continuous variables), few have explored tasks where the input dimension is very high (such as images - $110 \times 200 \times 3$ pixels as in our driving experiments). In order to effectively learn a policy that relies solely on high-dimensional input, we make the following improvements over the original GAIL framework.

It is well known that the traditional GAN objective suffers from vanishing gradient and mode collapse problems [24, 25]. We propose to use the Wasserstein GAN (WGAN [26]) technique to alleviate these problems and augment our objective function as follows:

$$\min_{\theta,\psi} \max_\omega \mathbb{E}_{\pi_\theta}[D_\omega(s,a)] - \mathbb{E}_{\pi_E}[D_\omega(s,a)] - \lambda_0 \eta(\pi_\theta) - \lambda_1 L_I(\pi_\theta, Q_\psi) - \lambda_2 H(\pi_\theta) \quad (5)$$

We note that this modification is especially important in our setting, where we want to model complex distributions over trajectories that can potentially have a large number of modes.

We also use several variance reduction techniques, including baselines [27] and replay buffers [28]. Besides the baseline, we have three models to update in the InfoGAIL framework, which are represented as neural networks: the discriminator network $D_\omega(s,a)$, the policy network $\pi_\theta(a|s,c)$, and the posterior estimation network $Q_\psi(c|s,a)$. We update $D_\omega$ using RMSprop (as suggested in the original WGAN paper), and update $Q_\psi$ and $\pi_\theta$ using Adam and TRPO respectively. We include the detailed training procedure in Appendix C. To speed up training, we initialize our policy from behavior cloning, as in [12].

Note that the discriminator network $D_\omega$ and the posterior approximation network $Q_\psi$ are treated as distinct networks, as opposed to the InfoGAN approach where they share the same network parameters until the final output layer. This is because the current WGAN training framework requires weight clipping and momentum-free optimization methods when training $D_\omega$. These changes would interfere with the training of an expressive $Q_\psi$ if $D_\omega$ and $Q_\psi$ share the same network parameters.

## 4 Experiments

We demonstrate the performance of our method by applying it first to a synthetic 2D example and then in a challenging driving domain where the agent is imitating driving behaviors from visual inputs. By conducting experiments on these two environments, we show that our learned policy $\pi_\theta$ can 1) imitate expert behaviors using high-dimensional inputs with only a small number of expert demonstrations, 2) cluster expert behaviors into different and semantically meaningful categories, and 3) reproduce different categories of behaviors by setting the high-level latent variables appropriately.

The driving experiments are conducted in the TORCS (The Open Source Racing Car Simulator, [15]) environment. The demonstrations are collected by manually driving along the race track, and show typical behaviors like staying within lanes, avoiding collisions and surpassing other cars. The policy accepts raw visual inputs as the only external inputs for the state, and produces a three-dimensional continuous action that consists of *steering*, *acceleration*, and *braking*. We assume that our policies are Gaussian distributions with fixed standard deviations, thus $H(\pi)$ is constant.

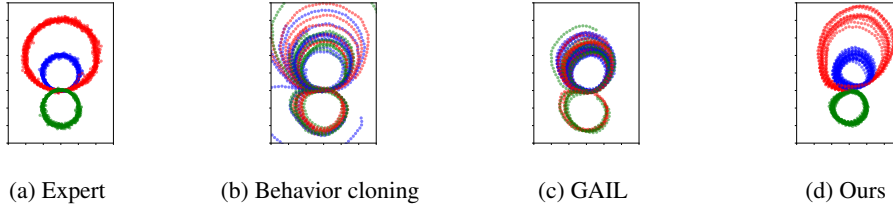

(a) Expert          (b) Behavior cloning          (c) GAIL          (d) Ours

Figure 1: **Learned trajectories in the synthetic 2D plane environment.** Each color denotes one specific latent code. Behavior cloning deviates from the expert demonstrations due to compounding errors. GAIL does produce circular trajectories but fails to capture the latent structure for it assumes that the demonstrations are generated from a single expert, and tries to learn an average policy. Our method (InfoGAIL) successfully distinguishes expert behaviors and imitates each mode accordingly (colors are ordered in accordance to the expert for visualization purposes, but are not identifiable).

## 4.1 Learning to Distinguish Trajectories

We demonstrate the effectiveness of InfoGAIL on a synthetic example. The environment is a 2D plane where the agent can move around freely at a constant velocity by selecting its direction $\mathbf{p}_t$ at (discrete) time $t$. For the agent, the observations at time $t$ are positions from $t-4$ to $t$. The (unlabeled) expert demonstrations contain three distinct modes, each generated with a stochastic expert policy that produces a circle-like trajectory (see Figure 1, panel a). The objective is to distinguish these three distinct modes and imitate the corresponding expert behavior. We consider three methods: behavior cloning, GAIL and InfoGAIL (details included in Appendix A). In particular, for all the experiments we assume the same architecture and that the latent code is a one-hot encoded vector with 3 dimensions and a uniform prior; only InfoGAIL regularizes the latent code. Figure 1 shows that the introduction of latent variables allows InfoGAIL to distinguish the three types of behavior and imitate each behavior successfully; the other two methods, however, fail to distinguish distinct modes. BC suffers from the compounding error problem and the learned policy tends to deviate from the expert trajectories; GAIL does learn to generate circular trajectories but it fails to separate different modes due to the lack of a mechanism that can explicitly account for the underlying structure.

In the rest of Section 4, we show how InfoGAIL can infer the latent structure of human decision-making in a driving domain. In particular, our agent only relies on visual inputs to sense the environment.

## 4.2 Utilizing Raw Visual Inputs via Transfer Learning

The high dimensional nature of visual inputs poses a significant challenges to learning a policy. Intuitively, the policy will have to simultaneously learn how to identify meaningful visual features, and how to leverage them to achieve the desired behavior using only a small number of expert demonstrations. Therefore, methods to mitigate the high sample complexity of the problem are crucial to success in this domain.

In this paper, we take a transfer learning approach. Features extracted using a CNN pre-trained on ImageNet contain high-level information about the input images, which can be adapted to new vision tasks via transfer learning [29]. However, it is not yet clear whether these relatively high-level features can be directly applied to tasks where perception and action are tightly interconnected; we demonstrate that this is possible through our experiments. We perform transfer learning by exploiting features from a pre-trained neural network that effectively convert raw images into relatively high-level information [30]. In particular, we use a Deep Residual Network [31] pre-trained on the ImageNet classification task [32] to obtain the visual features used as inputs for the policy network.

## 4.3 Network Structure

Our policy accepts certain auxiliary information as internal input to serve as a short-term memory. This auxiliary information can be accessed along with the raw visual inputs. In our experiments, the auxiliary information for the policy at time $t$ consists of the following: 1) **velocity** at time $t$, which is a three dimensional vector; 2) **actions** at time $t-1$ and $t-2$, which are both three dimensional vectors; 3) **damage** of the car, which is a real value. The auxiliary input has 10 dimensions in total.

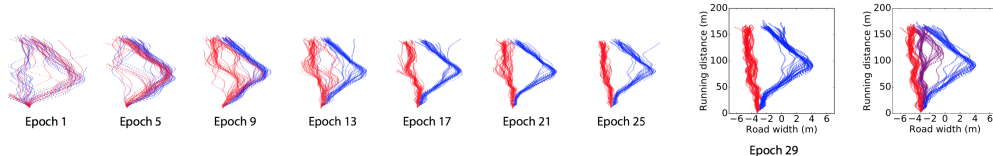

Figure 2: **Visualizing the training process of *turn*.** Here we show the trajectories of InfoGAIL at different stages of training. Blue and red indicate policies under different latent codes, which correspond to "turning from inner lane" and "turning from outer lane" respectively. The rightmost figure shows the trajectories under latent codes $[1, 0]$ (red), $[0, 1]$ (blue), and $[0.5, 0.5]$ (purple), which suggests that, to some extent, our method is able to generalize to cases previously unseen in the training data.

For the policy network, input visual features are passed through two convolutional layers, and then combined with the auxiliary information vector and (in the case of InfoGAIL) the latent code $c$. We parameterize the baseline as a network with the same architecture except for the final layer, which is just a scalar output that indicates the expected accumulated future rewards.

The discriminator $D_\omega$ accepts three elements as input: the input image, the auxiliary information, and the current action. The output is a score for the WGAN training objective, which is supposed to be lower for expert state-action pairs, and higher for generated ones. The posterior approximation network $Q_\psi$ adopts the same architecture as the discriminator, except that the output is a softmax over the discrete latent variables or a factored Gaussian over continuous latent variables. We include details of our architecture in Appendix B.

## 4.4 Interpretable Imitation Learning from Visual Demonstrations

In this experiment, we consider two subsets of human driving behaviors: *turn*, where the expert takes a turn using either the inside lane or the outside lane; and *pass*, where the expert passes another vehicle from either the left or the right. In both cases, the expert policy has two significant modes. Our goal is to have InfoGAIL capture these two separate modes from expert demonstrations in an unsupervised way.

We use a discrete latent code, which is a one-hot encoded vector with two possible states. For both settings, there are 80 expert trajectories in total, with 100 frames in each trajectory; our prior for the latent code is a uniform discrete distribution over the two states. The performance of a learned policy is quantified with two metrics: the *average distance* is determined by the distance traveled by the agent before a collision (and is bounded by the length of the simulation horizon), and *accuracy* is defined as the classification accuracy of the expert state-action pairs according to the latent code inferred with $Q_\psi$. We add constant reward at every time step as reward augmentation, which is used to encourage the car to "stay alive" as long as possible and can be regarded as another way of reducing collision and off-lane driving (as these will lead to the termination of that episode).

The average distance and sampled trajectories at different stages of training are shown in Figures 2 and 3 for *turn* and *pass* respectively. During the initial stages of training, the model does not distinguish the two modes and has a high chance of colliding and driving off-lane, due to the limitations of behavior cloning (which we used to initialize the policy). As training progresses, trajectories provided by the learned policy begin to diverge. Towards the end of training, the two types of trajectories are clearly distinguishable, with only a few exceptions. In *turn*, $[0, 1]$ corresponds to using the inside lane, while $[1, 0]$ corresponds to the outside lane. In *pass*, the two kinds of latent codes correspond to passing from right and left respectively. Meanwhile, the average distance of the rollouts steadily increases with more training.

Learning the two modes separately requires accurate inference of the latent code. To examine the accuracy of posterior inference, we select state-action pairs from the expert trajectories (where the state is represented as a concatenation of raw image and auxiliary variables) and obtain the corresponding latent code through $Q_\psi(c|s, a)$; see Table 1. Although we did not explicitly provide any label, our model is able to correctly distinguish over $81\%$ of the state-action pairs in *pass* (and almost all the pairs in *turn*, confirming the clear separation between generated trajectories with different latent codes in Figure 2).

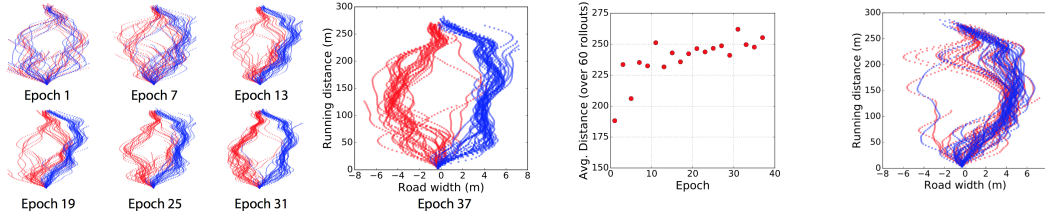

Figure 3: **Experimental results for *pass***. **Left**: Trajectories of InfoGAIL at different stages of training (epoch 1 to 37). Blue and red indicate policies using different latent code values, which correspond to passing from right or left. **Middle**: Traveled distance denotes the absolute distance from the start position, averaged over 60 rollouts of the InfoGAIL policy trained at different epochs. **Right**: Trajectories of *pass* produced by an agent trained on the original GAIL objective. Compared to InfoGAIL, GAIL fails to distinguish between different modes.

Table 1: Classification accuracies for *pass*.

| Method | Accuracy |
| --- | --- |
| Chance | 50% |
| K-means | 55.4% |
| PCA | 61.7% |
| **InfoGAIL (Ours)** | **81.9%** |
| SVM | 85.8% |
| **CNN** | **90.8%** |

Table 2: Average rollout distances.

| Method | Avg. rollout distance |
| --- | --- |
| Behavior Cloning | 701.83 |
| GAIL | 914.45 |
| InfoGAIL \ RB | 1031.13 |
| InfoGAIL \ RA | 1123.89 |
| InfoGAIL \ WGAN | 1177.72 |
| **InfoGAIL (Ours)** | **1226.68** |
| Human | 1203.51 |

For comparison, we also visualize the trajectories of *pass* for the original GAIL objective in Figure 3, where there is no mutual information regularization. GAIL learns the expert trajectories as a whole, and cannot distinguish the two modes in the expert policy.

Interestingly, instead of learning two separate trajectories, GAIL tries to fit the left trajectory by swinging the car suddenly to the left *after* it has surpassed the other car from the right. We believe this reflects a limitation in the discriminators. Since $D_\omega(s, a)$ only requires state-action pairs as input, the policy is only required to match most of the state-action pairs; matching each rollout in a whole with expert trajectories is not necessary. InfoGAIL with discrete latent codes can alleviate this problem by forcing the model to learn separate trajectories.

## 4.5 Ablation Experiments

We conduct a series of ablation experiments to demonstrate that our proposed improved optimization techniques in Section 3.2 and 3.3 are indeed crucial for learning an effective policy. Our policy drives a car on the race track along with other cars, whereas the human expert provides 20 trajectories with 500 frames each by trying to drive as fast as possible without collision. Reward augmentation is performed by adding a reward that encourages the car to drive faster. The performance of the policy is determined by the average distance. Here a longer average rollout distance indicates a better policy.

In our ablation experiments, we selectively remove some of the improved optimization methods from Section 3.2 and 3.3 (we do not use any latent code in these experiments). **InfoGAIL(Ours)** includes all the optimization techniques; **GAIL** excludes all the techniques; **InfoGAIL\WGAN** switches the WGAN objective with the GAN objective; **InfoGAIL\RA** removes reward augmentation; **InfoGAIL\RB** removes the replay buffer and only samples from the most recent rollouts; **Behavior Cloning** is the behavior cloning method and **Human** is the expert policy. Table 2 shows the average rollout distances of different policies. Our method is able to outperform the expert with the help of reward augmentation; policies without reward augmentation or WGANs perform slightly worse than the expert; removing the replay buffer causes the performance to deteriorate significantly due to increased variance in gradient estimation.

## 5 Related work

There are two major paradigms for vision-based driving systems [33]. *Mediated perception* is a two-step approach that first obtains scene information and then makes a driving decision [34–36]; *behavior reflex*, on the other hand, adopts a direct approach by mapping visual inputs to driving actions [37, 16]. Many of the current autonomous driving methods rely on the two-step approach, which requires hand-crafting features such as the detection of lane markings and cars [38, 33]. Our approach, on the other hand, attempts to learn these features directly from vision to actions. While mediated perception approaches are currently more prevalent, we believe that end-to-end learning methods are more scalable and may lead to better performance in the long run.

[39] introduce an end-to-end imitation learning framework that learns to drive entirely from visual information, and test their approach on real-world scenarios. However, their method uses behavior cloning by performing supervised learning over the state-action pairs, which is well-known to generalize poorly to more sophisticated tasks, such as changing lanes or passing vehicles. With the use of GAIL, our method can learn to perform these sophisticated operations easily. [40] performs end-to-end visual imitation learning in TORCS through DAgger [18], querying the reference policies during training, which in many cases is difficult.

Most imitation learning methods for end-to-end driving rely heavily on LIDAR-like inputs to obtain precise distance measurements [21, 41]. These inputs are not usually available to humans during driving. In particular, [41] applies GAIL to the task of modeling human driving behavior on highways. In contrast, our policy requires only *raw visual information* as external input, which in practice is all the information humans need in order to drive.

[42] and [9] have also introduced a pre-trained deep neural network to achieve better performance in imitation learning with relatively few demonstrations. Specifically, they introduce a pre-trained model to learn dense, incremental reward functions that are suitable for performing downstream reinforcement learning tasks, such as real-world robotic experiments. This is different from our approach, in that transfer learning is performed over the critic instead of the policy. It would be interesting to combine that reward with our approach through reward augmentation.

## 6 Conclusion

In this paper, we present a method to imitate complex behaviors while identifying salient latent factors of variation in the demonstrations. Discovering these latent factors does not require direct supervision beyond expert demonstrations, and the whole process can be trained directly with standard policy optimization algorithms. We also introduce several techniques to successfully perform imitation learning using visual inputs, including transfer learning and reward augmentation. Our experimental results in the TORCS simulator show that our methods can automatically distinguish certain behaviors in human driving, while learning a policy that can imitate and even outperform the human experts using visual information as the sole external input. We hope that our work can further inspire end-to-end learning approaches to autonomous driving under more realistic scenarios.

#### Acknowledgements

We thank Shengjia Zhao and Neal Jean for their assistance and advice. Toyota Research Institute (TRI) provided funds to assist the authors with their research but this article solely reflects the opinions and conclusions of its authors and not TRI or any other Toyota entity. This research was also supported by Intel Corporation, FLI and NSF grants 1651565, 1522054, 1733686.

## Footnotes

[1] A video showing the experimental results is available at `https://youtu.be/YtNPBAW6h5k`.

[2]For a fair comparison, we consider this form as our GAIL baseline in the experiments below.

[3][14] presents a proof for the lower bound.

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
