[Reviews · NeurIPS 2017]

Reviewer 1



Paper Summary: This paper focuses on using GANs for imitation learning using trajectories from an expert. The authors extend the GAIL (Generative Adversarial Imitation Learning) framework by including a term in the objective function to incorporate latent structure (similar to InfoGAN). The authors then proceed to show that using their framework, which they call InfoGAIL, they are able to learn interpretable latent structure when the expert policy has multiple modes and that in some setting this robustness allows them to outperform current methods. Paper Overview: The paper is generally well written. I appreciated that the authors first demon- started how the mechanism works on a toy 2D plane example before moving onto more complex driving simulation environment. This helped illustrate the core concepts of allowing the learned policy to be conditioned on a latent variable in a minimalistic setting before moving on to a more complex 3D driving simulation. My main concern with the paper is that, in integrating many different ideas, it was hard to distil the core contribution of the paper which, as I understood, was the inclusion of latent variables to learn modes within an overall policy when using GAIL. Major Comments: 1. In integrating many different ideas, it was difficult to distil the core contribution of this paper (the inclusion of latent variables to learn modes within a policy). As an illustration of this problem, In subsection 3.1 the authors refer to "InfoGAIL" in its simplest form, but the supplementary material they present a more complex algorithm which includes weight clipping, Wasserstein GAN objective as well as an optional term for reward augmentation and this is also called "InfoGAIL". I felt the paper could beneft from concretely defining InfoGAIL in its most simple form, and then when adding additional enhancements later and making it clear this is InfoGail with these extra enhancements. 2. Linked to my first point, I felt the paper would benefit from an algorithm box for the simplest form of InfoGAIL in the paper itself (not supplement- tary). This would add to the readability of the paper and allow a reader not too familiar with all the details of GANs to quickly grasp the main ideas of the algorithm. I'd suggest adding this before the section 4 on optimization. Then keeping the InfoGAIL with extensions in supplementary material (and renaming it accordingly there). 3. The authors show how InfoGAIL works in an idealized setting where the number of modes are known. It would be useful to see how InfoGAIL behaves when the number of modes is not known and the number of latent variables are set incorrectly. For example, in the 2D plane environment there are 3 modes. What happens if we use InfoGail but set the number of latent variables to 2, 4, 5, 10 etc. Can it still infer the modes of the expert policy? Is the method robust enough to handle these cases? Minor Comments: 1. subsection 4.1 refers to reward augmentation. But the experiment in sub- section 5.2 don't mention if reward augmentation is used to obtain results (reward augmentation is only mentioned for experiments in subsection 5.3). It's not clear if the authors use reward augmentation in this experiment and if so how it is defined. 2. The heading of the paper is "Inferring The Latent Structure of Human Decision-Making from Raw Visual Inputs". Does InfoGAIL only work in domains where we have raw visual inputs. Also we may assume the method would work if the expert trajectories do not come from a human but some other noisy source. The authors may want to revise the title to make the generality of their framework more clear.

Reviewer 2



The paper tackles the problem of learning by demonstration from raw visual inputs. As demonstrations may notably be provided by different experts, the authors propose to learn latent factors to disentangle the different demonstrations. The proposed technique builds on several previous works, namely GAIL (Generative Adversarial Imitation Learning) as general framework for learning from demonstration, InfoGAN to take into account latent factors, reward augmentation for taking into account a priori knowledge about the task, Wasserstein GAN as an extension of GAN to regression, use of a baseline and replay buffer for variance reduction, and finally initialization of the policy with behavior cloning. The method is evaluated in the TORCS environment on two tasks: turn and pass, where in both two different behaviors can be observed in the demonstrations. ** Strengths The authors tackle the important and hard problem of learning from demonstration directly from raw visual inputs. The proposed approach combines several previous techniques in a sound way as far as I understand. As demonstrated, it can distinguish two sets of trajectories in the TORCS environment. ** Weaknesses It seems the proposed method requires the a priori knowledge of the number of latent modes. How would the performance change if this number were incorrectly chosen? The clarity of the paper could be improved. Notably, it would help the reader if the overall objective function of the proposed were stated once. Otherwise it is a bit too implicit in my opinion. Besides, I think more emphasis should be put on the fact that the method needs auxiliary information (velocity, previous two actions and car damage). Currently I don’t think the method is fully end-to-end as claimed in the conclusion. Moreover I feel the abstract overstates the achievement of the proposed method. I wouldn’t qualify what the method achieves in the two tasks, turn and pass, as producing different driving styles. Other comments: - The authors should recall the definition of the notation E_{\pi_\theta}[f(s, a)] for a given function f. - In (2), Q -> Q_\psi - l.191: D -> D_\omega - l.232: output that indicates - l.317: it -> It

Reviewer 3



The overall problem is interesting, showcasing combination of several recently proposed methods (Wasserstein GAN, InfoGAIL etc). However, only two latent states considered in the experiment. Given that the experiment was only conducted in the driving domain, I'm not certain if the method would apply as well in other domains with more complicated latent structure